# MicroRNAs in Ruminants and Their Potential Role in Nutrition and Physiology

**DOI:** 10.3390/vetsci10010057

**Published:** 2023-01-14

**Authors:** Oyindamola Esther Ojo, Susanne Kreuzer-Redmer

**Affiliations:** Nutrigenomics Unit, Institute of Animal Nutrition and Functional Plant Compounds, University of Veterinary Medicine Vienna, 1210 Vienna, Austria

**Keywords:** microRNAs, ruminant nutrition, ruminant physiology, nutrigenomics, ruminant diseases, biomarkers, livestock diseases

## Abstract

**Simple Summary:**

The rising areas of nutrigenomics and feedomics determine the fashion of research in the field of veterinary medicine. In the present review, we summarize recent findings about how nutrition influences metabolic disorders and diseases through modifications in the activity and function of microRNAs. MicroRNAs perform essential functions in a variety of biological processes, including differentiation, cell proliferation, metabolism, development, and inflammation. Circulating microRNAs are being investigated as potential biomarkers of disease, susceptibility, and dietary exposure. Finally, we highlight the role of microRNAs as biomarkers in ruminant health and diseases, and discuss the difficulties of biomarker development in the transition from bench to clinical practice.

**Abstract:**

The knowledge of how diet choices, dietary supplements, and feed intake influence molecular mechanisms in ruminant nutrition and physiology to maintain ruminant health, is essential to attain. In the present review, we focus on the role of microRNAs in ruminant health and disease; additionally, we discuss the potential of circulating microRNAs as biomarkers of disease in ruminants and the state of technology for their detection, also considering the major difficulties in the transition of biomarker development from bench to clinical practice. MicroRNAs are an inexhaustible class of endogenous non-protein coding small RNAs of 18 to 25 nucleotides that target either the 3′ untranslated (UTR) or coding region of genes, ensuring a tight post-transcriptionally controlled regulation of gene expression. The development of new “omics” technologies facilitated a fresh perspective on the nutrition–to–gene relationship, incorporating more extensive data from molecular genetics, animal nutrition, and veterinary sciences. MicroRNAs might serve as important regulators of metabolic processes and may present the inter-phase between nutrition and gene regulation, controlled by the diet. The development of biomarkers holds the potential to revolutionize veterinary practice through faster disease detection, more accurate ruminant health monitoring, enhanced welfare, and increased productivity. Finally, we summarize the latest findings on how microRNAs function as biomarkers, how technological paradigms are reshaping this field of research, and how platforms are being used to identify novel biomarkers. Numerous studies have demonstrated a connection between circulating microRNAs and ruminant diseases such as mastitis, tuberculosis, foot-and-mouth disease, fasciolosis, and metabolic disorders. Therefore, the identification and analysis of a small number of microRNAs can provide crucial information about the stage of a disease, etiology, and prognosis.

## 1. Introduction

Feed regurgitation, chewing, salivation, and swallowing are described as rumination. Rumination reduces particle size, increasing microbial activity and making it easier for the digest to pass through the digestive tract. Ruminants digest their meals differently from non-ruminants due to their distinct gastrointestinal systems. Ruminating animals have several physiological traits that assist them in surviving in harsh environmental conditions. As a result, ruminants require different antibiotic-like compounds (mostly ionophores, lipophilic compounds) and direct-fed microbials than non-ruminants. These ruminant-specific products are thought to improve ruminal development, lactation, and production by influencing the generation of short chain fatty acids (SCFA) and microbial protein in the rumen to increase the provision of metabolizable energy and a balanced mix of amino acids [1]. The idea of nutritional value combines knowledge of the nutrients present and their availability, with considerations of the typical consumption rates, flavor, and the impact of the meal on the health of animals and the standard of animal products. Understanding ruminant responses to dietary components and other environmental factors elucidates the significance of the environment–gene expression interaction, including post transcriptional regulation through microRNAs (miRNAs). A better understanding of this interaction presents the opportunity to modulate metabolism, health, and diseases based on nutritional strategies.

Feeds are known to provide a conditioning environment that shapes the genome’s activity and the body’s physiology [2]. Direct studies of feeding methods associated with production characteristics have generally dominated the traditional approach in ruminant nutrition. However, this approach was unable to offer sufficient information regarding nutrient dynamics in the gastrointestinal tract (GIT), their impact on tissue, and molecular mechanisms of metabolism. As a result of advances in ruminant nutrition and physiology, molecular biology, high-throughput technologies, and bioinformatics databases, other approaches, such as nutrigenomics, epigenetics, metagenomics, metabolomics, transcriptomics, and proteomics have become more prominent. The environment modulates gene expression through modifications. One type of that modification of gene expression is epigenetics, and if epigenetics is thought of as a harmonizing process, many phenotypic differences can be simply explained [3]. Another mode of regulation of gene expression, influenceable through environmental factors, is facilitated post-transcriptionally through miRNAs.

Understanding gene regulation, including the effect of nutrition in altering noncoding regulatory RNA such as miRNAs, is one of the foundations of the biological and molecular underpinnings of how diet affects animal nutrition and phenotypic variation [4]. MiRNAs are single-stranded noncoding RNA molecules, which are 18–25 nucleotides (nts) long, that contribute to posttranscriptional gene regulation by binding (usually with imperfect complementarity) to the 3-UTR of a target messenger RNA (mRNA), resulting in translation degradation or inhibition [5]. Each miRNA is considered to have several targets, and multiple miRNAs can converge on a single mRNA, implying that these fascinating molecules play a significant regulatory role in metabolism and development [6]. Lin-4 and lethal-7 (let-7) were the first miRNAs discovered, and they were determined to be critical for developmental timing in Caenorhabditis elegans [7]. Following that, a mammalian let-7 family with twelve members expressed from eight separate loci (let-7a-1, -2, -3; let-7b; let-7c; let-7d; let-7e; let-7f-1, -2; let-7g; let-7i; miR-98) was discovered and characterized [8]. Even though they are found all over the genome, many let-7 miRNA family members are coordinated during development, and their regulation has been found to involve RNA-binding proteins, such as Lin28, which inhibits let-7 biogenesis. Since the discovery of the first miRNA (lin-4) in 1993 [9], advancements in next-generation sequencing (NGS, also referred as second-generation sequencing) and third-generation sequencing (TGS) technology have ushered in a new age and the ability to rapidly detect numerous classes of small RNA molecules, including miRNA, in various biological samples [10]. Repeated free-thaw cycles and long-term storage have been found to be stable for miRNAs in many biological samples, which make miRNAs a robust biological marker [11]. Various pathways, such as direct fusion, internalization, and receptor-mediated interactions, are hypothesized to be involved in delivery to destination cells and tissues. These functional miRNAs are supposed to use cellular machinery to regulate mRNA translation to protein once they have been delivered. Multiple sequencing systems (e.g., Illumina, Ion Torrent, and SOLiD) and bioinformatics data management skills enable in-depth miRNA sequencing (miRNA-Seq), allowing for the discovery of known and novel miRNAs [12,13], alterations [14], and possible functions [15].

An overview of the most recent research on ruminant miRNAs, their role in diet, and their potential as biomarkers for ruminant nutrition is given in this review. The physiological processes of the rumen receive no less consideration.

## 2. The Role of miRNAs in Ruminants

One of the most prosperous subspecies of terrestrial mammals is the ruminant family. They live in a wide range of diverse environments around the world and have a big impact on various ecosystems, industries including agriculture, leisure activities, and cultures [16]. The ability of this group of animals to survive and procreate on low-quality, low-protein, and high-fiber plant resources is a major factor in their success. Most of the various stakeholders believe that animal health is crucial to the production of livestock; however, there is a disagreement among consumers, farmers, and veterinarians over what constitutes an acceptable health state.

The term “animal health’’ lacks a precise scientific definition (ranging from the absence of disease to a broad definition of health as a state of unrestricted physical, physiological, and psychological well-being), as well as clear standards from which the condition of animals and the quality of their feed could be properly evaluated. One of the main goals of farmers is to assure the health and performance of their livestock through the adequate provision of suitable feed. By enhancing forage quality and creating rations with more balanced ratios of forage and grain, organic farming should attempt to boost the energy content of cattle diets to improve the efficiency of protein utilization and, as a result, reduce nitrogen loss to the environment [16]. Farm animals’ cell differentiation, biological development, and physiology are all significantly influenced by miRNAs. These processes include controlling muscle growth and hypertrophy, adipose tissue expansion, oocyte maturation, and early embryonic development [17]. Recent research has demonstrated the critical roles of miRNAs in sheep [18], goats [19], and bovine [20,21] rumen development, as well as the preservation of intestinal homeostasis. As a result, the miRNA expression profiles in the rumen, small intestine (duodenum and jejunum [22]), and large intestine (cecum and colon) of sheep and cattle have been identified. The study also revealed that some miRNAs are exclusively expressed in specific intestine segments, indicating that their roles may be constrained to the local microenvironment. In addition, taxonomic differences in how miRNAs regulate gene expression typically occur during the expression and processing stages [23]. In ruminants, the colon is a vital component of the hind gut’s digestive system. Cell wall polysaccharides, cellulose, and hemicellulose are fermented and used in large part by the colon in ruminants. According to Yan and colleagues’ study [23], from a total of 1572 miRNAs discovered in the colon tissues during the analysis of colon miRNA transcriptomes in preweaning and postweaning goats, 39 differentially expressed miRNAs and 88 highly expressed miRNAs were screened, and various functions of dynamic miRNAs in the regulatory system governing colon growth in goats were discovered [23]. In a study on cows from our group, the function of the rumen tissue miRNAome and transcriptome in relation to diet changes or the addition of a phytogenic feed additive was reported. We investigated how a phytogenic feed additive supplemented the diet transition from forage (FD) to high-grain (HG) feeding, and how that affected the role of miRNAs in the epithelial transcriptome. The study provided evidence that miRNAs have a direct role in the host’s responses to nutrition by identifying potential miRNA control mechanisms of gene expression during the switch from FD to HG feeding and phytogenic supplements [24]. In a parallel study from our research group, the presence of miRNAs in rumen fluid and the potential for miRNA-mediated cross-talk within the ruminal ecosystem were examined. The study hypothesized a potential role as a mechanism of interaction between the host and the ruminal microbiota, and suggested that this communication is bidirectional, with the microbiota influencing the host’s miRNA expression pattern and the host potentially helping to shape the gut bacterial profile through the production of specific miRNAs [25].

### Potential Regulatory Functions of miRNAs in Ruminant’s Milk

The ruminant mammary gland (MG) is a crucial organ responsible for producing milk for human infant and adult nutrition [26]. Not only nutrition, genetics, breed, disease pathogens, and other environmental factors, but also post transcriptional regulation of gene expression affects MG productivity. Lactation, one of the amazing outcomes of evolution, is a very dynamic and complicated process [27]. The growth of the MG and the production and release of milk are all parts of lactation. Cattle, buffaloes, goats, sheep, and camels provide almost all the world’s milk. Yaks, horses, reindeer, and donkeys are additional less frequent milk-producing animals. Each species’ prevalence and significance vary greatly between different geographical areas and nations. Feed, water, and climate are the three main factors that influence the dairy species retained. Market demand, dietary customs, and the socioeconomic makeup of each household are other variables that could impact the existence of a dairy species (e.g., poorer families tend to rely more on small ruminants). Since milk from dairy ruminants such as cows, goats, and sheep has been shown to have positive benefits on humans, extensive work has been carried out to increase milk production and improve its nutritional value [28]. Most infant formulas are based on cow milk proteins, and both cow and goat milk are commonly used as dairy products. Results show that adults absorb significant levels of milk-derived miRNAs from commercial pasteurized milk. Additionally, investigations have shown that milk exosomes can be incorporated into kidney, intestinal, intestinal cancer, and peripheral cells, as well as into macrophages and colon cancer cells [29]. According to Golan-Gerstl and colleagues, 95 percent of the miRNAs expressed in human milk are likewise expressed in goat and bovine milk, and pasteurization of bovine milk does not appear to eliminate miRNAs [30]. Additionally, 24 validated sites that were engaged in immunomodulatory actions were shared by conserved miRNAs [31]. Unsaturated fatty acid-rich feed added to the diet can be a useful strategy to boost milk’s health-promoting qualities; albeit, the impacts on the genes and pathways involved in these processes have not yet been fully and accurately described [32]. There are a number of measures, including nutrition, seasonal feed changes, and genomic variation, that can be utilized to improve the beneficial components in milk in ruminants because the process for the synthesis of milk fat is complex and subject to multifactorial regulation [33].

Bovine, caprine, and ovine species’ genetic variations of miRNAs expressed in the mammary gland or found in milk and localized in dairy quantitative trait loci (QTLs) were examined to find variations that might be the causes of dairy features. Using whole genome data to find miRNA genetic variants expressed in the mammary gland and localized in dairy QTLs, the study identifies miRNA genetic variations of interest in the context of dairy production [34].

A. Cattle

Proteins, lipids, and amino acids, as well as other bioactive substances such as hormones and cytokines, are all readily available in cow milk, which is also a good source of many other vital nutrients. Cow milk has been commercialized and regularly used by people for growth and health benefits due to its nutritional relevance. Li and colleagues describe the miRNA expression spectra of three milk fractions (fat, whey, and cells), contrast the milk fraction miRNome profiles with those of mammary gland tissue, and determine which milk fraction miRNome profile might be a better indicator of the miRNome profile of mammary gland tissue. Their findings demonstrated that the miRNAome of mammary gland tissue was accurately represented by miRNAs from milk fat. Top expressed miRNAs in milk were functionally annotated, and this revealed their crucial regulatory roles in mammary gland functions and perhaps to milk recipients [35]. Udder diseases, particularly mastitis brought on by bacterial infections, are significant issues for the dairy industry globally. Mastitis continues to be a major issue for the dairy industry globally, resulting in significant losses every year from reduced milk production (both quantity and quality), expensive treatments, and early animal culling, as well as having a significant impact on the development of antimicrobial resistance in cattle due to the widespread use of dry cow antibiotics [36]. Improved tools that can accurately detect early mammary inflammation in cattle are urgently needed given the relevance of early disease detection for minimizing the considerable financial and animal welfare implications of mastitis globally. Monitoring mammary gland health and spotting early inflammation require different methods. The most used method is somatic cell counting (SCC), which can be carried out in large quantities of milk or as individual milk samples, directly or indirectly through colorimetric quantification, frequently with the California Mastitis Test (CMT) [36]. Mammary epithelial cells release milk fat globules through a budding mechanism that encloses a crescent of the mammary epithelial cells cytoplasm in the plasma membrane [35]. Whether the miRNAs found in milk are specific to the mammary gland or come from the blood is the key question in determining the involvement of miRNAs in lactation. Tzelos and colleagues investigated the relationship between CMT scores (0 to 3), which were derived from many individual quarter milk samples (*n* = 236) taken from dairy cows at various lactations, and the levels of four inflammation-associated miRNAs (bta-miR-26a, bta-miR-142-5p, bta-miR-146a, and bta-miR-223). They confirmed that higher miRNA levels during lactation 1 than later lactations were responsible for the significant lactation number effect (P 0.01) for bta-miR-26a, bta-miR-142-5p, and bta-miR-146a. They also showed that bta-miR-223 and bta-miR-142-5p levels could accurately (100% sensitivity, >81% specificity) identify early inflammation. They stated that their findings offer further evidence of the usefulness of miRNAs as early diagnostic indicators of cow mastitis [36]. Using microarray and quantitative PCR analysis, Izumi and colleagues identified variations between colostrum (100 miRNAs) and mature milk (53 known miRNAs). They confirmed that some mRNA was present in cow’s milk, but that naturally occurring miRNA and mRNA in raw milk were resistant to acidic conditions and RNase treatment. Synthesized miRNA spiked in the raw milk whey were degraded [37]. Wang’s research team used transcriptome studies of mammary gland tissues from dairy cows during the high-protein/high-fat, low-protein/low-fat, or dry periods to investigate the molecular mechanisms governing milk secretion and quality in dairy cows. They discovered 25 core differentially expressed miRNAs (DE miRNAs) that were important for mammary gland growth and epithelial cell terminal differentiation during lactation, as compared to non-lactation. Their findings suggested that during mammary gland development, DE miRNAs might function as regulators of milk quality and milk secretion [38]. Xia and colleagues used miRNA and transcriptome data from the mammary epithelial cells of dairy cattle with high (H, 4.85%) and low milk fat percentages (L, 3.41%) during mid-lactation to screen and identify differentially expressed miRNAs, candidate genes, and co-regulatory pathways related to the metabolism of milk fat. In the co-expression networks of the dairy cattle mammary system, they discovered functional miRNAs and regulatory candidate genes involved in lipid metabolism (Table 1). This information advances our understanding of potential regulatory mechanisms of genetic elements and gene signaling networks involved in milk fat metabolism [39]. Since small non-coding RNAs have been linked to various phenotypes in bovine sperm, Werry and colleagues hypothesized that some differences in bull fertility may be reflected in the levels of various miRNAs in sperm. However, efforts to identify sperm-borne molecular biomarkers of male fertility have so far failed to identify a robust profile of expressed miRNAs related to fertility. The most abundant miRNAs in both populations were identified (miRs -34b-3p, -100-5p, -191-5p, -30d-4p, and -21-5p), and variations in both the total levels and particular patterns of isomiR expression were assessed. The findings offer a thorough description of the bovine sperm miRNAome and point to numerous potential roles in fertility [40]. To support the use of milk fat globules as a source of small non-coding RNAs to diagnose mastitis, their abundance from five cows before and after lipopolysaccharide (LPS) challenge was compared. Six miRNAs that are known to be regulated in the mammary gland during inflammation were also examined. The results showed that milk fat globules might be an easily accessible source of miRNAs that are possible biomarkers to detect early mastitis, and enable the application of a quick and efficient treatment. Four miRNAs (miR-494-3p, -148a-3p, -99a-5p, and -125b-5p) were variably abundant depending on the inflammatory status [41]. The Cui study team found 497 known miRNAs and 49 new ones using mRNA sequencing in the mammary glands of milking dairy cows. One of them, miRNA-71, was expressed differently in cows whose milk contained high and low levels of protein and fat [42]. A total of 21 differentially expressed genes can be referred to as targets for some of the 71 DE miRNAs based on their prior RNA sequencing data, suggesting that they may be crucial regulators of the milk protein and fat characteristics in dairy calves [43]. The miRNAomes of five essential metabolic tissues (rumen, duodenum, jejunum, liver, and mammary gland tissues) involved in protein synthesis and metabolism from 18 dairy cows fed high- and low-quality diets were studied to better understand the molecular regulatory mechanisms of milk protein production in dairy cows. There were 340, 338, 337, 330, and 328 miRNAs expressed in the rumen, duodenum, jejunum, liver, and mammary gland tissues, respectively. The findings indicated that miRNAs expressed in these tissues may play a part in controlling the transfer of amino acid for milk production downstream, which is a critical mechanism that may be related to low milk protein under poor forage feed [44]. When the mammary gland miRNomes of two dairy cow breeds, Holstein and Montbéliarde, with different mammogenic potentials that are related to differences in dairy performance, were compared, 754 distinct miRNAs were found in the mammary glands of Holstein (*n* = 5) and Montbéliarde (*n* = 6) midlactating cows. The outcome demonstrates variations in the mammary gland miRNomes of two dairy cattle breeds, and suggests that miRNAs may have a role in the flexibility of the mammary gland and the synthesis of milk components, both of which are connected to the quantity and composition of milk [45].

With Illumina RNA-sequencing, it was discovered that miRNAs have a regulatory role in the early development of the gastrointestinal tract (GIT). As a result of the findings, which included temporal and regional variations in miRNA expression as well as a connection between miRNA expression and the microbial population in the GIT during early life, there is now additional support for the theory that host–microbial interactions regulate gut development through a different mechanism [20]. Based on the variations in meat quality attributes and 90 differently expressed mRNAs, an integrated study of miRNA and mRNA expression profiling was carried out between bulls and steers, and 18 DE miRNAs were discovered. The findings offered compelling proof of the molecular genetic regulation and gene interactions in cattle [46].

B. Goats

Goat milk is typically made into cheese in Mediterranean nations and Latin America; in Africa and South Asia, it is typically drunk raw or acidified. One of the most significant livestock animals is the goat (Caprahircus) [47]. Numerous studies on goats have looked at the impact of various feeding methods on milk fat content; nevertheless, the physiological underpinnings of this reaction are still not well understood.

A study that examined the relationship between differentially expressed miRNAs in goat mammary tissue and the fatty acid composition of goat milk found that levels of miR-183, targeting MST1 (Macrophage Stimulating 1), were positively associated with the milk’s fatty acid content [48]. The MST1 gene is targeted by miR-183 in the cytoplasm of goat mammary epithelial cells, which results in an inhibition of milk fat metabolism. One of the most crucial aspects of the nutritional quality of goat milk is its lipid composition. For instance, the findings of the study by Ollier and colleagues revealed that whole intact rapeseeds or sunflower oil in high-forage or high-concentrate diets affected milk yield and composition, but not the mammary mRNA expression of the important genes involved in lactose (for example, α-lactalbumin), protein (for example, β-casein), and lipid metabolism (e.g., lipoprotein, lipase) [49]. The response to dietary interventions did not appear to be mediated by changes in the mRNA abundance of genes encoding essential lipogenic enzymes and the associated transcription factors, according to [50] on lactating goats fed a supplement of sunflower seed oil. Thirty highly expressed miRNAs, including miR-103, whose expression correlates with lactation, were found by high-throughput sequencing in the mammary gland of lactating goats. This study’s conclusions provided new insight about the roles of miR-103 and the molecular processes that control milk fat synthesis [51]. Diets containing extruded linseed alone or in conjunction with fish oil in lactating goats exhibited effects on mRNA connected to protein metabolism and transport pathways rather than lipid metabolism, as well as a significant alteration in the FA composition of milk [52]. MiR-25 mimics in goat mammary epithelial cells lowered the expression of genes involved in lipid metabolism, which was inversely correlated with milk production at various phases of lactation. The study’s findings revealed the role of the miR-25/PGC-1beta regulatory axis during lactation and suggested that miR-25 may be involved in lipid metabolism [53] (Table 1). A total of 1487 unique miRNAs, including 45 novel miRNA candidates and 1442 known and conserved miRNAs with 378 differentially expressed and 758 co-expressed miRNAs, were found between early and late lactation. The study’s findings suggested that miRNAs may be crucial to early and late lactation throughout the development of the dairy goat mammary gland, which will help researchers better understand how genes regulate mammary gland development and lactation [54]. The goat genome was sequenced, and 487 miRNAs were identified. The greatest miRNA clusters were discovered on chromosome 21 [55]. Overall, 131 novel and 300 conserved miRNAs were identified after analyzing goat MG tissues during early lactation using the Illumina-Solexa high-throughput sequencing method [56]. Additionally, 346 conserved and 95 novel miRNAs were discovered in goat MG tissues from dry off, and peak lactation does use the same technique (Illumina-Solexa sequencing).

C. Sheep

Most sheep in the world live in developing nations. In colder regions they are even more common than goats. Sheep farming includes a variety of products that can be produced, including milk, meat, skin, fiber, and dung, although most small-scale farmers in developing nations grow sheep for meat or for sale as livestock in local markets. Milk production and lactation duration have not significantly increased because of genetic selection in dairy sheep. Awassi, East Friesian, and Lacaune are sheep breeds used for dairy products [47].

In a study with different fat-tailed sheep breeds, 155 DE miRNAs, including 78 up-regulated and 77 down-regulated miRNAs, were found between the tail fat tissue of Hu sheep (short-fat-tailed) and Tibetan sheep (short-thin-tailed) using miRNA-Seq. The findings might offer a useful theoretical framework for research into the molecular processes behind sheep tail adipogenesis [57]. Using RNA sequencing and cell-level validation (an error-based approach to design and optimization) is crucial when dealing with a complicated process such as NGS. The degree of validation and quality control required for specific process steps can be determined by carefully considering the likelihood of potential failure spots. It also helps in troubleshooting any errors and validating changes made to various test system components. The role of miRNA in the deposition of intramuscular fat (IMF) was investigated, and 59 DE-miRNAs were discovered between 2-month-old (Mth-2) and 12 month-old (Mth-12) Aohan fine-wool sheep (AFWS). In an effort to enhance the quality of sheep meat, the study identified lists of miRNAs linked to intramuscular lipid deposition in sheep and their prospective targets [58]. A study compared the microstructures and the miRNA expression profile of mammary gland (MG) tissues at peak lactation in small-tailed Han and Gansu Alpine Merino sheep, with various milk production attributes. Eighteen of the one hundred and twenty-four mature miRNAs produced were differentially expressed between the two breeds. The findings indicated that the functions of miRNAs in the growth and lactation of MG in sheep can be improved. The results also indicated that the targeted genes of differentially expressed miRNAs were mainly involved in metabolic pathways and signaling pathways related to MG development, milk protein, and fat synthesis [59] (Table 1).

**Table 1 vetsci-10-00057-t001:** MiRNAs involved in the regulation processes of mammary lipid metabolism of ruminants.

miRNAs	Targeted Genes and Pathways	Regulating Functions	References
miR-103	PKAN3, AMPKα pathway	Accelerates de novo synthesis of fatty acids/Unsaturated or saturated fatty acids ratio	[51,60]
miR-224	ALOX15, PTGS1, ACADM	Milk fat metabolism/Increased apoptosis rate	[61]
miR-221	FASN, NR1H3, ACSL1	Droplet of lipid formation	[62]
miR-24-3p	MEN1	Regulates synthesis of milk proteins	[63]
miR-24	ACACA, TIP47, GPAM	Droplet of fat formulation, concentration of fatty acid, synthesis fatty acid	[64]
miR-486	PTEN	Regulates phosphoacyl alcohol signal transduction	[65]
miR-124a	PECR	Metabolizes fatty acid	[66]
miR-135a	PPLR	Regulates prolactin secretion	[67]
miR-106b	ABCA1	Accumulates triglycerides and cholesterol in epithelial cells of the mammary gland	[68]
miR-145	INSIG1	Stimulates the production of milk fat	[69]

## 3. MiRNAs Involved in Disease and Health in Ruminants

Ruminant diseases cause significant financial losses worldwide by increasing mortality and decreasing productivity in dairy herds. External and internal parasites, mastitis, and other production-related diseases usually do not kill the animal, but always make the system less effective. Diseases can have an impact on dairy productivity by lowering milk production, reducing fertility, delaying puberty, lowering milk quality, and reducing feed conversion. Health risks associated with dairy animal diseases could potentially spread to humans (e.g., tuberculosis, brucellosis). Small-scale dairy farming in poor nations is vulnerable to numerous health concerns. Numerous factors contribute to this, including poor understanding of disease prevention, treatment, and control; a high prevalence of infections; and the price, accessibility, or suitability of animal health services. A small-scale dairy producer with few resources may experience significant household economic effects from the death of even one animal due to disease. Small-scale dairy farmers typically do not invest much in animal health, particularly in terms of disease prevention. Depending on their physical and physiological traits, various dairy species and breeds have various health requirements. By choosing dairy animals that are compatible with the local environment, animal health and welfare issues may be considerably decreased. The ability of the dairy animal to adapt to the climate, graze on the available resources, and fend off endemic diseases and local parasites are of relevance. While dairy animals kept in extensive systems are more vulnerable to parasite diseases, those kept in intensive systems are more susceptible to agents that cause transmissible disease. Animals may be more susceptible to endemic diseases when they are introduced to a new area because they lack developed immunity [47].

A. Mastitis

The most expensive disease in dairy production is mastitis, an inflammation of the mammary glands, which is mostly brought on by a wide variety of bacteria that can be classified as infectious and environmental bacteria. Mastitis, particularly the sub-clinical variety, is the most prevalent production-related disease in cattle, sheep, and goats that are extensively managed for the purpose of producing meat or milk [70] (Figure 1).

The negative impact of mastitis includes not only effects on the economy, but also important effects on the animals’ health and welfare. Food safety (food-borne diseases) and the quality of dairy products (such as cheese) are relevant perspectives as well, since milk from affected animals may include pathogenic bacteria and have altered compositions that are undesirable to the dairy sector [71,72]. *Staphylococcus aureus*, *Streptococcus agalactiae*, *Strep.* spp., *Corynebacterium bovis* (summer mastitis of heifers, dry cows, and beef breeds), coliform agents (*E. coli* and *Klebsiella pneumonia*), and *Mycoplasma* spp., especially *M*. *bovis,* are the most prevalent bovine mastitis pathogens (California mastitis). Many of these pathogens can lead to significant systemic dysfunction, including fever and anorexia, as well as acute and chronic mastitis. Mastitis in cattle exhibits comparable signs and can be treated in a manner like for small ruminants [73].

Some miRNAs may have important functions during mastitis. In the mammary tissue of cows with mastitis, different levels of expression of several miRNA target genes, including interleukin-8 (*IL-8*) and granulocyte-macrophage colony-stimulating factor (GM-CSF), were discovered [74]. In addition, Li et al. discovered that miR-31 and miR-205 were down-regulated in mammary epithelial tissue infected with *S. aureus* relative to healthy controls, whereas miR-223 was increased (Table 2) [75]. The primary mastitis pathogen is *S. aureus*; however, there have been a few investigations on the dynamic changes in miRNA expression in peripheral blood during the onset and development of mastitis in dairy cows infected with *S. aureus*. The expression level of miRNAs that are linked to mastitis in dairy cows is very sensitive to changes in the pathogenic bacterium type, and dosage and period of infections. For the molecular diagnosis and biological therapy of mastitis, it is crucial to comprehend the changes in miRNA during the pathogenesis [76]. In a recent study, dairy cows’ mammary gland tissues were infected with 5 mL of 10^5^ CFU/mL of S. aureus to prepare a mastitis cow model. Finding biomarkers for early identification of S. aureus-infected mastitis in dairy cow peripheral blood is therefore crucial. The findings provided a new basis for investigation into molecular diagnostics and biological therapy for S. aureus-infected mastitis in dairy cows [76]. It is necessary to comprehend the control of immunological mechanisms during mastitis, particularly the role of miRNAs, as the genetic basis of mastitis is yet unknown. Using next-generation sequencing, the miRNA profile of the parenchyma was found to change during mastitis, with its profile depending on the type of pathogen. It was found to be changed depending on whether the glands were infected with coagulase-positive staphylococci (CoPS) or coagulase-negative staphylococci (CoNS). They came to the conclusion that their discoveries regarding the impact of the discovered miRNAs on the etiology of mastitis represent a new step in understanding the disease’s molecular mechanisms and may enable more effective prevention and treatment, as well as functional studies on the function of miRNAs in the regulation of molecular pathways related to bacterial infection [74].

B. Tuberculosis

As a zoonotic disease, tuberculosis (TB) can spread from animals to humans. Cattle and Sheep tuberculosis is typically caused by *Mycobacterium bovis* (*M. bovis)* or *M. avium. Mycobacteria* are pleomorphic, aerobic, nonmotile, nonspore-forming bacilli. Goat cases have been linked to *M. tuberculosis*, *M. avium*, or *M. bovis*. *M. bovis* has a diverse range of hosts and infects cattle with tuberculosis. Animal TB, especially in cattle, has a devastating influence on animal productivity and public health because of its importance. Trade in live animals and their products is severely restricted because of TB in cattle, resulting in significant economic losses [77]. The prevention of the spread of the virus and the reduction of transmission would be greatly aided by the early detection of *Mycobacterium avium subspecies paratubercolosis* (MAP) affected animals. Both in human and veterinary medicine, circulating miRNA profiles and gene expression have been suggested as potential indicators of disease. As a result, gene expression and associated miRNA levels were examined in cows that had tested positive for MAP using ELISA and culture in attempts to find potential biomarkers to help in MAP infection diagnosis (Table 2). The findings indicated that MAP infection has an impact on miRNA expression and that miRNAs are crucial in controlling how the host reacts to infection. The gene expression and miRNA profiles may serve as biomarkers of infection, and an earlier method of diagnosing MAP infection than the currently available ELISA-based diagnostic techniques [78]. A study describes a first next-generation sequencing approach to temporally profile miRNA expression in primary bovine alveolar macrophages (AMs) following *Mycobacterium bovis* (*M. bovis*) infection. *Mycobacterium bovis*, the causative agent of bovine tuberculosis, spreads through the air and is taken up by alveolar macrophages in the lung. According to the research, miRNAs are crucial in regulating the intricate interaction between *M. bovis* survival tactics and the host immune response [79].

C. Brucellosis

An economically significant zoonotic disease that can impact humans, domesticated animals, and even wildlife, is brucellosis. Gram-negative bacteria of the genus *Brucella* are responsible for the disease. Four of the six identified classical *Brucella* species are thought to be harmful to humans. The most frequent cause of human brucellosis is *Brucella melitensis*, which primarily infects goats and sheep, while *B. abortus*, which primarily infects cattle, buffalo, elk, yaks, and camels, is the second most frequent source of infection in humans. Animal brucellosis mostly affects females’ genitalia and causes abortions, though orchitis or joint manifestations have occasionally been reported [80].

In a recent study, the serum miRNA signature linked to brucellosis in water buffaloes was characterized, and the potential for using the miRNAs as biomarkers in vaginal secretions was examined. Dysregulated miRNAs in blood serum and vaginal fluids were confirmed using RT-qPCR, and miRNA expression profiles in Brucella-positive and Brucella-negative blood sera were assessed using next-generation sequencing. The outcomes showed possible indicators for Brucella infection in water buffaloes and gave an overview of miRNA expression levels (Table 2). A better understanding of the molecular processes underlying Brucella infection and host immune response is possible if additional functional and mechanistic studies of these miRNAs are considered [81].

D. Foot and Mouth

Globally, one of the most significant and urgent livestock diseases is foot and mouth disease (FMD) (Figure 1). In endemic areas of the world, its yearly economic impact on observable production losses and vaccine costs is estimated to range from US$6.5 to US$21 billion. FMD is a highly contagious disease that affects animals with cloven hooves and is brought on by the FMD virus (FMDV), a type of *Aphthovirus* found in the family *Picornaviridae* [82]. Cattle, pigs, sheep, goats, and water buffalo (*Bubalus bubalis*) are among the domesticated species that are susceptible to FMD. The presence of subclinical disease types makes FMD control particularly challenging. In a proof-of-concept work utilizing miRNA PCR array plates, the relative abundance of 169 miRNAs was determined in bovine serum obtained at three different stages of FMDV infection. A different circulating miRNA profile from animals that had recovered from infection was induced by subclinical FMDV persistence. While bta-miR-31 was most abundant during FMDV persistence, bta-miR-17-5p was most abundant during acute infection (Table 2). These results imply that non-coding regulatory RNAs are involved in FMDV infection of cattle. Future research will outline the unique contributions of the identified miRNAs to FMDV replication, as well as whether this miRNA profile is shared by all FMDV serotypes and whether it may be used to build innovative diagnostic applications [83]. Cattle, a natural host for FMDV, were used to study the host miRNA response after FMDV infection. Early in the infection process, a significant alteration in serum miRNA expression was found. The data show that changes in miRNA expression take place during early pathogenesis, and the identification of potential miRNA target genes may aid in unravelling the molecular processes involved in the interaction between the virus and the host, which may be helpful in the development of therapeutic approaches [84].

E. Fascioliasis

Some species of leaf-like digenetic trematodes in the genus Fasciola are responsible for the old food-borne, but neglected, zoonotic disease known as fascioliasis (Figure 1). Following ingestion of the parasite, fascioliasis has an asymptomatic incubation phase, which is followed by an acute and a chronic clinical phase. The juvenile worms breach the peritoneum and intestinal wall to initiate the acute phase of the *Fasciola* infection [85]. They subsequently move on to the liver surface and the bile ducts. A substantial number of nations around the world are affected by fascioliasis, with Latin America and the Middle East reporting the highest burdens. According to estimates, *F. hepatica* infects about 300 million cattle and 250 million sheep worldwide, and along with *F. gigantica*, it is thought to be responsible for an annual economic loss of USD 3 billion [85]. To establish a baseline for the prevalence of *Fasciola gigantica* infestation in cattle butchered at the Minna metropolitan abattoir in the Chanchaga Local Government area of Niger State, Nigeria, a 90-day study was conducted by Osinowo et al. This investigation proved that *Fasciola gigantica* was present in animals butchered in Minna Metropolis. Cattle should only be allowed to graze in regions with fewer snail infestations, especially those near rivers and streams [86].

A total of 121 host circulating miRNAs were differentially expressed in this study’s two groups of twenty-four (8–10-month-old) buffaloes, of which 44 miRNAs were up-regulated and 77 miRNAs were significantly down-regulated. Four parasite-derived miRNAs were found in the sera of *F. gigantica*-infected buffaloes, and the host circulating miRNAs were dysregulated in the buffalo sera during infection (Table 2). The outcome will enhance circulating parasite-derived miRNAs as diagnostic targets of parasite infection, and contribute to the understanding of the molecular mechanisms underlying host–parasite interactions [87].

F. Peste des Petit

A virus from the family *Paramyxoviridae*, genus *Morbillivirus*, is the culprit behind the peste des petits ruminants. Prior to 2016, this virus’s official name was Peste des petits ruminants virus (PPRV); it was then changed to *Small ruminant morbillivirus* (SRM). However, professionals in the field still refer to it as PPRV. The virus is a pleomorphic particle with a ribonucleoprotein core and RNA genome enclosed by a lipoprotein membrane. The genome is a single stranded, negative polarity, negative sense RNA that is roughly 16 kilobases (kb) long [88]. Africa, the Middle East, Central Asia, and East Asia are all affected by the endemic small ruminant plague. PPRV lineage IV has lately become widely distributed in Africa and Asia (such as China, Nepal, India, and Pakistan) (from the north to Tanzania). In Turkey, epidemics occur often with the most recent being in 2011–2016. In Europe, Bulgaria was where an incident was first noted in June 2018 [88]. PPRV-specific antibodies have been employed by numerous researchers in a variety of assays and tests for the detection of virus antigen in tissue, swabs, conjunctival smears, and formalin-fixed tissues. For the time being, the “gold standard” for PPR diagnosis is virus isolation. Primary sheep and cattle cells, as well as well-established cell lines such as Marmoset B-lymphoblastoid-B95a cells and Vero (African green monkey kidney) cells, can all be used to isolate and culture the PPRV in vitro [89]. To determine the function of differentially expressed miRNA (DEmiRNA) in PPR virus (PPRV)-infected lung and spleen tissues of sheep and goats, miRNAs were sequenced, and proteomics data were obtained. According to the research, PPRV-induced miR-21-3p, miR-320a, and miR-363 may work together to down-regulate many immune response genes in the lung and spleen of sheep, enhancing viral pathogenesis (Table 2). The fact that the PPRV–Izatnagar/94 isolate causes a stronger host response in goats than in sheep provides vital information on the molecular pathogenesis of PPR [90]. To identify the cellular miRNA expression profile in goat peripheral blood mononuclear cells (PBMC) infected with the commonly used vaccine strain Nigeria 75/1 for mass immunization campaigns against Peste des petits ruminants, the deep sequencing technique was applied. The findings of this work serve as an important foundation for further research into the functions of miRNA in PPRV replication and pathogenesis [91].

**Table 2 vetsci-10-00057-t002:** MiRNAs involved in important diseases of ruminant animals.

Potential miRNA Biomarkers	Diseases	Pathogens	Sample Tissue	References
miR-2339, miR-21-3p, miR-423-5p, miR-499, miR-92a, miR-193a-3p, miR-23a, miR-99b, miR-21-3p, miR-193a-3p, miR-365-3p, miR-30c, and miR-30b-5 pmiR-31, miR-205, miR223	Mastitis	*Staphylococcus aureus*	BMEC	[75,92]
miR-144, miR-451 and miR-7863		*Escherichia coli* and *Staphylococcus aureus*	BMEC	[93]
miR-21, miR-146a, miR-155, miR-222, and miR-383		*Streptococcus agalactiae*	Milk	[94]
let-7i, miR-21, miR-27, miR-99b, miR-146, miR-147, miR-155 and miR-223		California mastitis test (CMT)	Milk	[94]
miR-17-5p, miR-31 and miR-1281	Foot and Mouth disease	Foot and Mouth disease virus	Serum	[83]
miR-21-5p, miR-101, miR-126-3p, miR-145, miR-197, bta-miR-223			Serum	[84]
bta-miR-142-5p, bta-miR-146a and bta-miR-423-3p	Tuberculosis	*Mycobacterium bovis*	Lung	[79]
mir-19b, mir-19b-2, mir-1271, mir-100, mir-301a, mir-32, mir-6517 and mir-7857			Blood	[78]
miR-21-3p, miR-1246, miR-27a-5p, miR-760-3p, miR-320a and miR-363	Peste des petits ruminant’s virus infection	Peste des petits ruminant’s virus	Spleen and lung	[90]
miR-204-3p, miR-338-3p, miR-30b-3p, miR-199a-5p, miR-199a-3p and miR-1			Peripheral blood mononuclear cells	[91]
miR-let-7f, miR-151, miR-30e, miR-191, miR-150 and miR-339b	Brucellosis	*Brucella abortus*	Serum	[81]
miR-87, miR-71	Fascioliasis	*Fasciola gigantica*	Serum	[87]

## 4. Development of miRNAs as Biomarkers of Diet, Nutritional Status, and Their Potential for Therapeutic Use in Ruminants

Biomarkers are biological molecules used to comprehend a physiological process or identify an abnormal process or a disease. For the management of livestock diseases, miRNA can be employed as biomarkers that can be used for therapeutic, prognostic, or diagnostic purposes. They are also known as molecular markers, biochemical markers, or signature molecules [95]. A miRNA biomarker refers to miRNA that is generated or enriched particularly in each tissue, and whose circulating levels may indicate pathological or physiological changes in said tissue [96]. The choice of biomarkers in nutrigenomics must consider alterations in homoeostasis that reflect the relationship between nutritional diet and health or disease. However, recent studies cited in this review suggested that miRNAs from plasma, leucocytes, serum, and feces might be relevant biomarkers to quantify the physiological effects of dietary or intervention lifestyle studies (Table 2). MiRNAs from plasma or serum, PBMC, and feces may be useful biomarkers to measure the physiological effects of dietary or lifestyle intervention studies, according to current studies described in this review [97]. Most small RNA biomarker discovery studies employ a high-throughput profiling strategy, such as sequencing or PCR arrays, to find candidate sequences linked to a certain physiological or pathological condition. Cell-free biofluid samples can either be examined whole (such as plasma), or isolated fractions containing short RNAs can be examined (e.g., exosomal or lipoprotein fractions). Unfractionated biofluid samples may be more effective for achieving full screening of miRNA populations because studies have demonstrated that different fractions in plasma contain different miRNA sequences [98]. Most often, individual RT-qPCR assays are used for validation, allowing some control over enzymatic inhibition and sample contamination. The validated biomarkers can subsequently be tested in larger subject cohorts and/or have their function examined with in vitro or in vivo utilizing techniques, such as luciferase reporter assays and gain-of-function or loss-of-function methodologies. In a recent study, it was discovered that different diets could affect miRNA expression of bovine serum, and their potential influence on immunity was discovered as well. The findings suggested that one could use effective dietary strategies to interfere with the physiological state of animals [99].

Circulating miRNAs have been utilized as biomarkers for a variety of diseases and physiological conditions in mice and humans [100,101,102,103,104]; research on domestic animals, particularly ruminants, is limited. The discovery of deregulated miRNAs and their targets opens the possibility of developing therapeutic, prognostic, and diagnostic approaches for veterinary diseases. MiRNAs are naturally produced into body fluids by cells, where they remain in comparatively stable protein or lipid complexes, and are simple to measure (Figure 2). This gives the option to employ miRNAs as non-invasive indicators of tissue function linked to a variety of physiological states (such as pregnancy) and disorders. This is complemented with the fact that some miRNAs are tissue or developmental stage specific (e.g., neoplasia, cardiovascular disease, osteoarthritis, sepsis). Utilizing illumina small-RNA sequencing, plasma samples were taken from eight non-pregnant heifers on pregnancy days to determine the potential of circulating miRNAs as indicators of early pregnancy in cattle. MiR-26a was discovered by genome-wide analysis to be a possible circulating biomarker of early pregnancy, and plasma miRNA populations linked to early pregnancy in cattle were successfully characterized [105]. When the effects of age and genetic background on the expression patterns of 306 plasma miRNAs were examined in 18 animals, it was discovered that these factors were related to the attributes of health and production in dairy cows. Circulating miRNAs may serve as helpful markers for dairy cows to help with better health, welfare, and production outcomes, according to the study’s findings [106]. Six non-pregnant Holstein–Friesian cross cows were used in the analysis of the miRNA profiles in plasma and cell fractions of blood to find tissue-derived miRNAs that may be useful as indicators of tissue function in this major food animal species. An important factor in the context of post-partum negative energy balance in dairy cows, the study discovered miR802, a circulating miRNA that had not previously been identified in cattle and that may regulate insulin sensitivity and lipid metabolism. As a result, it may provide a specific biomarker of liver function [96]. The fact that circulating miRNA profiles alter in response to diseases, bacterial, or viral infections [81,83,107], and physiological states [108,109] (such as pregnancy), demonstrates the suitability of circulating miRNAs as biomarkers for tracking various physical situations in animals. It is becoming clearer that dietary feed components play a crucial function in nutrition and health, as well as in the regulation of miRNAs. Since a variety of bioactive feed ingredients affect how miRNAs are made, it stands to reason that some of these elements may also affect how diseases are susceptible to developing, intensifying, and progressing [2]. Cattle, a natural host for foot and mouth disease virus (FMDV), were used to study the host miRNA response following FMDV infection (Table 2). At the earliest stages of infection, a considerable change in serum miRNA expression was found. The study found that early pathogenesis is accompanied by changes in miRNA expression, and that finding potential miRNA target genes may aid in unravelling the molecular processes underlying virus–host interaction and, consequently, in the development of therapeutic approaches [84].

## 5. Future Recommendations and Open Research Suggestions

Biomarker studies, as discussed in this review, have been carried out in many species and concentrate on the diagnosis and monitoring of a wide range of diseases. The majority of the found biomarkers have not yet been transformed into useful diagnostic tests or commercially viable products, even though many are thought to be promising. For a biomarker to reach clinical usage, the entire set of essential validation processes has to be completed or published. The dairy cow still presents one of the most intensively farmed animals worldwide. Dairy cows with high milk yields have been genetically chosen for this trait, making them prone to diseases such mastitis, tuberculosis, foot and mouth, brucellosis, fasciolosis, and rumen acidosis. Therefore, the development of biomarkers for the early diagnosis of these diseases delineates an important mission of contemporary dairy research. Emerging new technologies, such as systems biology and omics approaches, have been widely used for the identification of ruminant biomarkers; however, the development of faster methods enabling a higher analytical sensitivity is inevitable. The future success of miRNAs as biomarkers in this area, and the realization of the improvements in ruminant production efficiency and welfare that relate to it depend on recognizing and then overcoming these limitations. Research funding continues to be a significant barrier to the development of miRNA biomarkers because these markers must be translated into clinical practice, which takes more time and money, and requires additional extensive studies with a large sample size for validation to ensure that they are truly able to predict the clinical outcome. An illustration of this is the use of urinary estrogens to detect pregnancy in giant pandas, where researchers have been analyzing estrogenic metabolites as markers of pregnancy, and viable cub development studies working towards this biomarker began already ten years ago [110,111].

Biomarkers still need to be linked into automated systems, platforms, and technologies before farmers and veterinarians can use them in the field. For example, even though pregnancy-associated glycoproteins can be thought of as particular biomarkers for fetal wellbeing and pregnancy diagnosis, such biomarkers have not yet been applied at farm level. In our opinion, successful solutions for enhancing the health, performance, and wellbeing of ruminants would necessitate interdisciplinary cooperation of basic scientists, farmers, consultants, veterinarians, and bioinformaticians.

## 6. Conclusions

MiRNAs have undeniable potential as biomarkers for the control of livestock diseases, and play important regulatory functions in ruminant nutrition and physiology. The use of miRNAs as biomarkers in ruminant nutrition and physiology, however, requires more investigation to identify the biomarkers for ruminant welfare and create accessible, affordable, and sensitive analytical or non-invasive approaches to assess them. Nevertheless, a substantial rise in miRNA research has been seen in recent years due to the urgent need to control ruminant diseases and disorders, such as sub-acute ruminal acidosis. MiRNA research benefits greatly from more potent computing resources, improved statistical techniques, and decreased sequencing and genotyping costs. Thus, we anticipate that miRNA biomarkers will eventually be created and used as effective instruments to control ruminant diseases.

## Figures and Tables

**Figure 1 vetsci-10-00057-f001:**
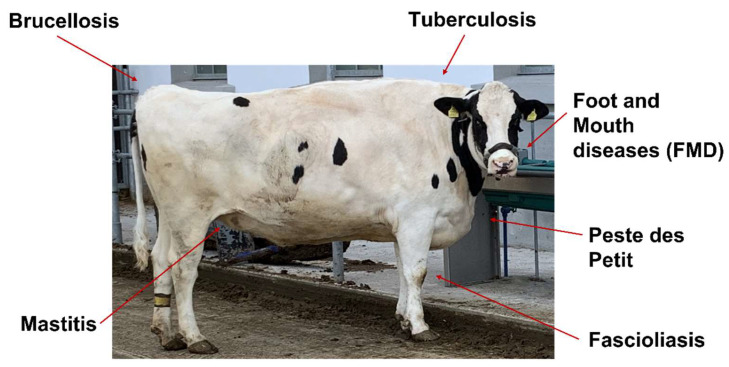
Important infectious diseases in Cows that are known to be at least partially regulated through miRNA action.

**Figure 2 vetsci-10-00057-f002:**
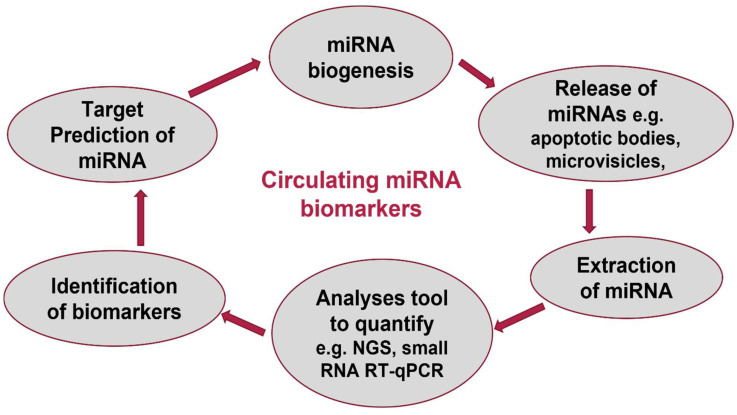
Circulating miRNAs as biomarkers.

## Data Availability

Not applicable.

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
