# Peer review of "MicroRNAs in Ruminants and Their Potential Role in Nutrition and Physiology"

_vetsci, 2023, doi:10.3390/vetsci10010057_

Round 1
Reviewer 1 Report
The present review manuscript tried to summarize the previous findings in related to the interaction between miRNA and nutrition in ruminant production, which is a novel point. While instead of focusing on this topic, this literature review spent great length to introduce some points which have been well known. Another important part which is missing in this review was that no suggestive comments were given.
Author Response
Reviewer 1
The present review manuscript tried to summarize the previous findings in related to the interaction between miRNA and nutrition in ruminant production, which is a novel point. While instead of focusing on this topic, this literature review spent great length to introduce some points which have been well known. Another important part which is missing in this review was that no suggestive comments were given.
Response: The authors appreciate the thorough feedback. To provide farmers, veterinarians, and researchers with suggestions and recommendations, we have a Section 5.
Reviewer 2 Report
General Comments
This is a well-written, very timely, and informative review article that provides both the broad biological concept for the novice while providing detailed biological technical information for expert researchers in this field. I commend the authors for this work.
I have detailed some minor technical edits that may need to be applied. The only major recommendation I have is that both the Simple Summary and Abstract are actually background/context statements, rather than an abbreviated summary of the review’s findings. Perhaps, it may be necessary to re-write them.
Specific Comments/queries
Simple Summary
Line 10 - Is omics a word; should omics be “omics” instead?,,,,
Line 21 – comma after “role”
Main Body
Line 68 “…other approaches, such as…” instead of “…other approaches, as…”?
Line 79 Sometimes miRNA is spelled out at the beginning of sentences, sometimes not. Here, it is not (MiRNA). Also throughout the manuscript, the abbreviation miRNAs is used and sometimes it is not (“microRNA” is used).
Line 89 – which is it, “coordinated” or “regulated”?
Lines 95-98 – Is there a citation to support this critical (miRNA stability) understanding?
Line 121 – Are the authors saying that animals living on organic farms are living in sub-optimal conditions? If not, reword this sentence.
Line 142 - 144 – need citation.
Line 303 – Need a period after “acidified”.
Line 313 “β-casein” instead of “b-casein”?
Line 369 – Spell-out “18”?
Line 408 – Be more definitive/specific than “serious issue”.
Line 437 – “.. dosage and period …..”
Table 1 – “Table 1.” Instead of “Table 1:”?
Figure 2 – The terms “miRNA’ and “microRNAs” are used in the circles of this figure. Should a single term (miRNA or microRNAs) be used instead?
Author Response
Reviewer 2
General Comments
This is a well-written, very timely, and informative review article that provides both the broad biological concept for the novice while providing detailed biological technical information for expert researchers in this field. I commend the authors for this work.
I have detailed some minor technical edits that may need to be applied. The only major recommendation I have is that both the Simple Summary and Abstract are actually background/context statements, rather than an abbreviated summary of the review’s findings. Perhaps, it may be necessary to re-write them.
Response: The authors are very thankful for the detailed comments. We have addressed all the typos and read the manuscript all over again. We also re-wrote our simple summary and abstract following the recommendation.
Specific Comments/queries
Line 10 - Is omics a word; should omics be “omics” instead?,,,,
Response: Line 10 – Yes, you are right! A biological science branch of study that ends in -omics, such as genomics, transcriptomics, proteomics, or metabolomics, is referred to as an omics. When referring to the study subjects in these domains, such as the genome, proteome, transcriptome, or metabolome, the suffix -ome is employed. As a result, we have already made the necessary adjustments in the manuscript.
Line 21 – comma after “role”
R: Addressed as suggested
Line 68 “…other approaches, such as…” instead of “…other approaches, as…”?
R: Addressed
Line 79 Sometimes miRNA is spelled out at the beginning of sentences, sometimes not. Here, it is not (MiRNA). Also throughout the manuscript, the abbreviation miRNAs is used and sometimes it is not (“microRNA” is used).
R: Thanks for pointing that out! The word MicroRNAs is introduced at the beginning of the manuscript (L57), and miRNAs/miRNA should be used throughout the manuscript for uniformity. Therefore, we corrected were necessary.
Line 89 – which is it, “coordinated” or “regulated”?
R: Coordinated is indeed the right word, corrected in line 88
Lines 95-98 – Is there a citation to support this critical (miRNA stability) understanding?
R: A citation was added as suggested. Citation 11: Matias-Garcia PR, Wilson R, Mussack V, Reischl E, Waldenberger M, Gieger C, et al. Impact of long-term storage and freeze-thawing on eight circulating microRNAs in plasma samples. PLoS One. 2020;15(1):e0227648.
Line 121 – Are the authors saying that animals living on organic farms are living in sub-optimal conditions? If not, reword this sentence.
R: Line 121 – Rephrased in line 117 – 122
Line 142 - 144 – need citation.
R: A citation was added as suggested. Citation 23: Yan Q, Tian L, Chen W, Kang J, Tang S, Tan Z. Developmental Alterations of Colonic microRNA Profiles Imply Potential Biological Functions in Kid Goats. Animals (Basel). 2022 Jun 14;12(12):1533.
Line 303 – Need a period after “acidified”.
R: Corrected in line 301.
Line 313 “β-casein” instead of “b-casein”?
R: The correction has been done in line 314.
Line 369 – Spell-out “18”?
R: Modified accordingly in line 366.
Line 408 – Be more definitive/specific than “serious issue”.
R: Addressed accordingly in line 405 -406.
Line 437 – “.. dosage and period …..”
R: Modified in line 438.
Table 1 – “Table 1.” Instead of “Table 1:”?
R: Modified as suggested
Figure 2 – The terms “miRNA’ and “microRNAs” are used in the circles of this figure. Should a single term (miRNA or microRNAs) be used instead?
R: We made the necessary adjustments in Figure 2.
Reviewer 3 Report
MicroRNAs in ruminants and its potential role in Nutrition 2 and Physiology
Observations:
This review paper is an excellent work. The authors showed a fascinating, essential, and innovative topic. Furthermore, they offer sufficient references to support the necessity to identify miRNAs as markets for diseases (mastitis, tuberculosis, etc.) in ruminants. Congratulations to the authors.
Author Response
Reviewer 3
Comments and Suggestions for Authors:
MicroRNAs in ruminants and its potential role in Nutrition 2 and Physiology
Observations:
This review paper is an excellent work. The authors showed a fascinating, essential, and innovative topic. Furthermore, they offer sufficient references to support the necessity to identify miRNAs as markets for diseases (mastitis, tuberculosis, etc.) in ruminants. Congratulations to the authors.
Response: The authors are very thankful for the positive comments.